# Association between Chronotype and Sleep Quality among Chinese College Students: The Role of Bedtime Procrastination and Sleep Hygiene Awareness

**DOI:** 10.3390/ijerph20010197

**Published:** 2022-12-23

**Authors:** Yingying Zhu, Jiahao Huang, Minqi Yang

**Affiliations:** 1Key Research Base of Humanities and Social Sciences of the Ministry of Education, Academy of Psychology and Behavior, Tianjin Normal University, Tianjin 300387, China; 2Faculty of Psychology, Tianjin Normal University, Tianjin 300387, China; 3Tianjin Social Science Laboratory of Students’ Mental Development and Learning, Tianjin 300387, China; 4School of Education, Zhengzhou University, Zhengzhou 450001, China; 5School of Marxism, Zhengzhou University, Zhengzhou 450001, China

**Keywords:** chronotype, bedtime procrastination, sleep quality, sleep hygiene awareness, college students

## Abstract

Chronotype and sleep quality have been shown to play significant roles in influencing people’s physical and mental health. The current study focuses on examining the relationship between chronotype and sleep quality among Chinese college students and exploring the mediating role of bedtime procrastination and the moderating role of sleep hygiene awareness. A sample of 2822 college students (female = 71.4%) aged between 17 and 29 years (M = 19.77, SD = 1.41) were included and completed the measures of the Pittsburgh Sleep Quality Index (PSQI), Morningness–Eveningness Questionnaire (MEQ), Bedtime Procrastination Scale (BPS) and Sleep Hygiene Awareness Scale (SHAS). The results showed that evening-type students reported the lowest sleep quality and highest levels of bedtime procrastination. In contrast, the highest sleep quality and lowest levels of bedtime procrastination were shown by morning-type, exhibiting the neither-type students’ intermediate chronotype. Bedtime procrastination partially mediated the relationship between chronotype and sleep quality. Furthermore, sleep hygiene awareness moderated the direct effect of chronotype on sleep quality and the effect of chronotype in the path from chronotype to bedtime procrastination. Specifically, higher levels of sleep hygiene awareness could buffer the adverse effect of chronotype on self-reported sleep quality but bolstered the negative effect of chronotype on bedtime procrastination. Our results suggest that individuals with an evening preference are inclined to postpone their bedtime and further experience poorer sleep quality at night. Sleep hygiene awareness may serve as a protective factor for poor nocturnal sleep. Overall, the findings highlight the importance of reducing bedtime procrastination and improving sleep hygiene awareness in the interventions designed to help college students to own a better sleep quality, especially for those with evening chronotypes.

## 1. Introduction

Chronotype, or morningness/eveningness, is a unique personal biological clock system that is determined by daytime activities and bedtime preferences. Researchers generally distinguish three main chronotypes: morning (“larks”), intermediate and evening (“owls”) [1]. Specifically, morning-type individuals tend to wake up early in the morning and achieve their peak mental and physical performance at this time, whereas evening-types, the so-called ‘owls’, demonstrate the opposite characteristics: wake up and go to bed late in the evening and perform best at the end of the day, often in the late hours of the evening [2]. A growing number of studies have shown the impacts of chronotype on individuals’ physical and psychological health. For instance, evening-types were found to be associated with lower levels of subjective well-being [3], a poorer lifestyle [4], and a greater incidence of sleep problems [5], as well as negative moods such as depression [6], anger [7], confusion [8] and fatigue [9] than morning-types. Changes in the amount of melatonin secretion in accordance with the length of sunlight exposure may account for these differences [10].

Relations between chronotype and individuals’ sleep quality have attracted much more attention during past decades. Sleep quality refers to the subjective perception of the duration and effectiveness of sleep [11]. Research has shown that many sleep problems were commonly observed, especially among evening types. Individuals with a circadian preference for evening types, for example, showed a higher prevalence of sleep insufficiency, poor sleep quality, and irregular sleep patterns than morning and intermediate types [12,13]. According to the sleep–wake regulation theory [14], a human’s sleep–wake cycle is regulated by sleep homeostasis and circadian processes, and circadian dysregulation between individual internal circadian rhythm and daily activities can easily lead to sleep disruptions and complaints. Findings from cross-sectional studies have shown that individuals with a circadian preference for evening activities (evening-type) showed a higher prevalence of sleep problems (e.g., decreased sleep quality, problems with sleep onset and maintenance) compared to morning or intermediate types [9,15]. A recent empirical research also demonstrated that sleep quality would be an important mediator in the relationship between circadian chronotype and mental health [16]. Additionally, from the perspective of social jetlag, people with evening chronotypes are more prone to suffer from daytime impaired cognitive functions and sleep disturbances due to the conflict between their internal biological clock and social/working schedules [17]. Therefore, with reference to previous findings, we proposed the hypothesis that chronotype may predict sleep quality among Chinese college students (H1).

Bedtime procrastination is usually defined as failing to go to bed as intended, without having external reasons for doing so [18]. A close relationship between bedtime procrastination and circadian chronotype has been detected in some prior studies [19,20,21,22]. Specifically, people with evening chronotypes showed a greater tendency to procrastinate at bedtimes compared to morning- and intermediate-types. Díaz-Morales and colleagues [23] have indicated that the eveningness was more likely to have more erroneous beliefs about the importance of sleep than their morning-type peers, and the unawareness of how important sleep is for their health, well-being, and daytime functioning may facilitate themselves to experience several problems with sleep behavior regulation, such as the failure control of bedtime behaviors. A recent work using a sample of Polish adolescents further confirmed the predictive role of chronotype on bedtime procrastination [20]; that is, evening-type adolescents showed a higher probability of going to bed later than morning-types, and in addition, the evening types were significantly related to lower levels of autonomous motivation for regulating sleep behaviors, which may, in turn, increase their tendency to delay bedtime and frequencies of experienced insufficient sleep. Some researchers also demonstrated from a time perspective that morningness tended to have superior general temporal adaptation and was more likely to consider the future consequences of their behavior and act more strategically; hence, these time perspective dimensions may prevent them from developing inappropriate behaviors or life habits [24]. In contrast, eveningness generally tended to be more impulsive and risky, with a preference for seeking immediate pleasures, even at the expense of mental health [25]. Hence, it is plausible to infer that circadian chronotype may associate with individuals’ bedtime procrastination.

Nowadays, delaying bedtimes is a very prevalent phenomenon in the Chinese population, especially among college students [26]. A study conducted on 414 Chinese college students revealed that more than 52% students had the habit of putting off their bedtime, and the reason for staying up late was mainly related to playing with mobile phones before going to bed. According to the “gratification theory” of the internet, smartphone users can experience feelings of satisfaction and happiness [27], whereas the pursuit of temporary satisfaction from using smartphones enables individuals to rather ‘satisfice’ their sleep time instead of going to bed on schedule, finally leading to the prolonged sleep latency and delayed sleep onset. In a recent cross-sectional study, Zhang and colleagues demonstrated the predictive effect of bedtime procrastination on individuals’ sleep quality, and also revealed the negative impact of addictive behaviors, including smartphone addiction on individuals’ subjective sleep quality via increasing bedtime procrastination behaviors [28]. Moreover, some empirical research found that this particular type of procrastination was also associated with people’s self-control capacities [29,30]. That is, people with a low self-control capacity were more likely to go to bed later than they intended, and this behavior might in turn have a significant negative impact on their nocturnal sleep, like decreased sleep quality and sleep insufficiency. Similarly, You and colleagues [31] demonstrated from Chinese college students that those with low self-regulation skills were prone to delaying their bedtime, which further gave rise to students’ poorer sleep at night. Overall, based on the literature reviewed above, given that chronotype is closely related with bedtime procrastination, and bedtime procrastination may be regarded as an indicator of sleep status, we hypothesized that bedtime procrastination would play a mediating role in the effect of chronotype on college students’ subjective sleep quality (H2).

Sleep hygiene refers to individual practices or routines that promote better sleep [32]. The principles of sleep hygiene may vary from one source to another, but there is a list of generally accepted rules, such as not going to bed hungry or thirsty, proper sleep scheduling (e.g., without daytime naps, consistent bed and wake times), avoiding stimulant substances (e.g., caffeine, cigarettes) and strenuous exercise within a short time before bedtime, as well as maintaining a comfortable sleep environment (e.g., the suitability of room light, noise, and temperature) [33,34]. Sleep hygiene awareness is about the knowledge or awareness of sleep hygiene people acquire [35]. It has been found that individuals with better sleep hygiene awareness usually had regular sleep habits and reported higher levels of sleep quality and sleep sufficiency; in contrast, those with poor sleep hygiene awareness were more likely to develop bad sleep behaviors, such as, delaying bedtime and irregular routines [36], and were prone to suffer from sleep and emotional disorders [37,38]. Some cross-sectional studies investigating the relationship between circadian typology and sleep hygiene knowledge indicated that evening-type subjects showed more wrong beliefs about sleep than that of the morning- and neither-types [39,40], further supporting previous findings that poor sleep health knowledge may lead to a greater incidence in the evening-type of irregular sleep habits and sleep problems [36,41]. Additionally, theories about sleep-interfering and sleep-interpreting processes have advocated that sleep-related behaviors and the perceptions and cognitions for these behaviors play a significant role in regulating nocturnal sleep [42]. A growing number of clinical and experimental studies also confirmed the effectiveness of using some cognitive methods, such as changing one’s mistaken beliefs, improving sleep hygiene awareness, and revising false attributions concerning the causes and effects of poor sleep to treat patients with insomnia and circadian rhythm disorders [43,44,45]. Hence, we would assume that sleep hygiene awareness might be a plausible moderator in the relationship between chronotype and sleep quality within the mediating role of bedtime procrastination (H3).

Taken together, the present study aimed to hypothesize a moderated mediating model with bedtime procrastination as a mediating variable and sleep hygiene awareness as a moderating variable to systematically explore the effects of bedtime procrastination and sleep hygiene awareness in the process of chronotype affecting sleep quality among Chinese college students, and meanwhile providing more empirical evidence for the intervention and prevention of poor sleep-related behaviors and problems in young adults.

## 2. Materials and Methods

### 2.1. Participants and Procedure

Participants were recruited from two universities in Tianjin and Shangqiu, China, using a convenience sampling method. A total of 2822 college students aged between 17 and 29 years (M_age_ = 19.77, SD_age_ = 1.41, 71.4% female) participated in this research. Most of the students were undergraduates, freshmen, sophomore, junior and senior students account for 30.7%, 39.2%, 22.3%, and 6.4%, respectively, and the others are postgraduate students (1.4%). After informing the participants of the purpose of our survey and obtaining their informed consent, our researchers delivered the questionnaires to them by sending them the QR code (Quick Response code) or a link to our questionnaires. All participants were assured of the anonymity and confidentiality of their answers. This study was reviewed and approved by the ethics committee at the authors’ institution and was thus performed in accordance with ethical standards laid down in the 1964 Declaration of Helsinki and its later amendments.

### 2.2. Measures

#### 2.2.1. Chronotype

The Morningness–Eveningness Questionnaire (MEQ) was used to measure whether an individual is a morning, evening, or intermediate (neutral) type. The questionnaire consists of 19 items that concern the preferred timing of certain activities, such as bed times, sleep timing and tests [46]. The Chinese version of the scale has demonstrated satisfactory construct validity and internal reliability [47]. Items have a response scale with four or five options (items 1, 2, 10, 17, 18 have four options and items 3–9, 11–16, 19 have five options). Sample items are “How alert do you feel during the first half after having woken in the mornings” and “At what time in the evening do you feel tired and as a result in need of sleep”. The total score ranges from 16 to 86. Scores between 59–86 identify participants as morning types, between 42–58 as intermediate types, and between 16 and 41 as evening types. The Cronbach’s α coefficient in this study was 0.71.

#### 2.2.2. Bedtime Procrastination

The Chinese version [48] of the Bedtime Procrastination Scale (BPS) [18] was used to measure participants’ bedtime procrastination behaviors. The BPS consists of 9 items with a 5-Likert point scale, ranging from 1 = “never” to 5 = “always”. Sample items are “I go to bed later than I had intended” and “Often I am still doing other things when it is time to go to bed”. Items 1, 4, 5, 6 and 8 were forward scored, and items 2, 3, 7 and 9 were inversely scored. Higher scores indicate more reported bedtime procrastination behaviors. In the present study, BPS Cronbach’s α coefficient was 0.76.

#### 2.2.3. Sleep Quality

Sleep quality was measured with the Pittsburgh Sleep Quality Index (PSQI) which is a self-rated questionnaire that assesses participants’ sleep quality and sleep disturbance during a 1-month period [49]. The Chinese version of the PSQI has well-established validity and reliability [50]. This scale has a total of 19 items, which is composed of seven subcategories, including subjective sleep quality, sleep disturbances, habitual sleep efficiency, sleep latency, sleep duration, hypnotic medication use and daytime dysfunction. Sample items are “During the past month, how would you rate your sleep quality overall” and “During the past month, how often have you taken medicine to help you sleep”. Each dimension of the PSQI scale is scored between 0 and 3, and the total score ranges from 0 to 21. Higher scores indicate worse sleep quality, and the respondents would be recognized as poor sleepers if they get a PSQI global score of 7. In the present study, the Cronbach’s α coefficient was 0.81.

#### 2.2.4. Sleep Hygiene Awareness

The Chinese version of the Sleep Hygiene Awareness Scale (SHAS) was used to assess participants’ sleep hygiene awareness [51]. The scale has been shown to have good validity and reliability in previous studies [31,52]. This scale consists of 13 items rated on a 7-point Likert scale (1 represents “strongly beneficial to sleep”, 7 represents “strongly disruptive to sleep”). Items 8, 10, 11 and 12 were inversely scored. Sample items are “whether regular use of sleep medications is beneficial to sleep, disruptive to sleep or have no effect” and “waking up at the same time every day is beneficial to sleep, disruptive to sleep or have no effect”. The total scores were calculated by adding up all the scores on each item. The obtained scores range from 13 to 91 points, with a higher score indicating more sleep hygiene knowledge. The Cronbach’s α coefficient in this study was 0.60.

### 2.3. Statistical Analysis

Data analyses were processed using SPSS 21.0 (IBM Corp., Armonk, NY, USA) and PROCESS 3.4. First, the comparisons between bedtime procrastination, sleep hygiene awareness and sleep quality according to the MEQ scores were tested using analysis of covariance (ANCOVA) with gender and age as covariates. Next, Pearson’s correlations were performed to determine the links between variables of interest. The third step was to test the mediation model and moderated mediation model using the SPSS macro PROCESS version 3.4 (model 4 and model 59) suggested by Hayes with the bootstrapping method [5000 samples]. A significant indirect effect was considered to be established when the 95% bias-corrected CIs did not contain 0 [53]. Finally, to further understand the nature of the moderation effect, conditional direct and indirect effects (“simple slopes”) were estimated by using the “pick-a-point” approach [54], with the sample plus and minus 1 SD from the mean representing “high” and “low” sleep hygiene awareness.

## 3. Results

### 3.1. Sociodemographic Data

A total of 808 male students (28.6%) and 2014 female students (71.4%) were included in the study. Out of the 2822 participants (M ± SD = 19.77 ± 1.41 years), 765 (27.1%, 255 males, 510 females) were identified as morning types (MT, mean age: 19.79 ± 1.41 years), 1880 (66.6%, 520 males, 1360 females) as intermediate types (IT, mean age: 19.76 ± 1.40 years) and 177 (6.3%, 33 males, 144 females) as evening types (ET, mean age: 19.90 ± 1.44 years) (Table 1). Table 2 demonstrates the comparisons of the values of BPS, PSQI and SHA among the chronotype groups by ANCOVA. Results showed that the main effects of chronotype on bedtime procrastination (*F*(2, 2817) = 254.67, *p* < 0.001) and sleep quality (*F*(2, 2817) = 116.37, *p* < 0.001) were both statistically significant. Further post-hoc comparisons by Bonferroni’s test showed that compared with morning- and intermediate-types, evening-types showed significantly higher scores in bedtime procrastination and sleep quality, as well as its seven components (*p*s < 0.01). No significant differences in sleep hygiene awareness were observed between these three chronotypes (*F*(2, 2817) = 0.16, *p* = 0.849).

### 3.2. Pearson’s Correlation Results

Correlation analysis was performed to examine the associations between chronotype, bedtime procrastination, sleep quality and sleep hygiene awareness. The results are presented in Table 3. Specifically, chronotype was strongly negatively correlated with bedtime procrastination (*r* = −0.47, *p* < 0.001) and sleep quality (*r* = −0.33, *p* < 0.001). Bedtime procrastination was strongly positively correlated with sleep quality (*r* = 0.42, *p* < 0.001), and a weak correlation was also found between sleep hygiene awareness and sleep quality (*r* = 0.06, *p* < 0.01). In addition, gender showed significant correlations with the main variables, including chronotype, bedtime procrastination, sleep hygiene awareness and sleep quality (*p* < 0.01), whereas age showed no significant correlations (except sleep quality) with chronotype, bedtime procrastination and sleep hygiene awareness (*p* > 0.05).

### 3.3. Mediation Analysis

The model 4 and bootstrap methods from the SPSS macro PROCESS 3.4 were used to test the significance of the mediating effect with chronotype as the independent variable, bedtime procrastination as the mediating variable and sleep quality as the dependent variable, gender and age as controlled variables. The results are shown in Table 4 and Figure 1. Chronotype was significantly and negatively associated with sleep quality (*β* = −0.33, *p* < 0.001) and bedtime procrastination (*β* = −0.46, *p* < 0.001) which was positively related to sleep quality (*β* = 0.34, *p* < 0.001). The direct effect of chronotype on sleep quality (*β* = −0.17, *p* < 0.001) and the indirect effect of chronotype on sleep quality via bedtime procrastination (*β* = −0.16, 95% CI [−0.18, −0.14]) were also significant. These findings indicated that bedtime procrastination partially mediated the relationship between chronotype and sleep quality among Chinese college students.

### 3.4. Moderation Analysis

The macro PROCESS (model 59) was adopted to test the moderated mediation model, in which chronotype was included as the independent variable, bedtime procrastination as the mediator and sleep quality as the dependent variable. Gender and age were set as covariates. The results showed that chronotype was negatively associated with bedtime procrastination (*β* = −0.45, *t* = −27.15, 95% CI [−0.49, −0.42]) and sleep quality (*β* = −0.17, *t* = −8.87, 95% CI [−0.21, −0.13]). Bedtime procrastination was positively related to sleep quality (*β* = 0.35, *t* = 17.30, 95% CI [0.31, 0.39]). Meanwhile, the chronotype × sleep hygiene awareness interaction effects on bedtime procrastination (*β* = −0.11, *t* = −6.57, 95% CI [−0.14, −0.08]) and sleep quality (*β* = 0.05, *t* = 2.73, 95% CI [0.02, 0.09]) were significant (Table 5), which indicated that the direct path between chronotype and sleep quality, and the path from chronotype to bedtime procrastination were moderated by sleep hygiene awareness. However, the moderating effect of sleep hygiene awareness in the path from bedtime procrastination to sleep quality was not significant (*β* = 0.01, *t* = 0.36, 95% CI [−0.03, 0.05]).

To better explain the moderated mediation model, based on the average score of sleep hygiene awareness plus or minus one standard deviation, the participants were divided into a high sleep hygiene awareness group (M + SD) and a low sleep hygiene awareness group (M − SD). Through simple slope analysis, we further examined the effects of chronotype on bedtime procrastination and sleep quality under different levels of sleep hygiene awareness. Results showed that the effect of chronotype on sleep quality among students with high levels of sleep hygiene awareness (*β*_simple_ = −0.12, *t* = −4.35, *p* < 0.001) was relatively smaller than that among the students with low levels of sleep hygiene awareness (*β*_simple_ = −0.22, *t* = −8.05, *p* < 0.001) (Figure 2). Whereas, the effect of chronotype on bedtime procrastination among students with high levels of sleep hygiene awareness (*β*_simple_ = −0.56, *t* = −24.93, *p* < 0.001) was larger than that among students with low levels of sleep hygiene awareness (*β*_simple_ = −0.34, *t* = −13.83, *p* < 0.001) (Figure 3).

## 4. Discussion

In this cross-sectional study on Chinese college students, we were the first to explore the relations between chronotype, bedtime procrastination and sleep quality, as well as the possible mechanisms. Our results indicated that chronotype negatively predicted sleep quality and bedtime procrastination among college students. Bedtime procrastination was significantly positively associated with students’ poor sleep quality. Meanwhile, bedtime procrastination played a mediating role in the relationship between chronotype and sleep quality, while the mediation effect was moderated by sleep hygiene awareness. Specifically, sleep hygiene awareness moderated the direct path between chronotype and sleep quality, as well as the first half path of chronotype via bedtime procrastination on the quality of sleep. These findings proposed a moderated mediation model that, on the one hand, could deepen our previous understanding of the relationship between chronotype and subjective sleep quality [1,13], and on the other, may provide some empirical evidence for the prevention and interventions on reducing bedtime procrastination behaviors and improving nighttime sleep among college students.

Increasing evidence showed that evening types were more likely to suffer from lower sleep quality and daytime sleepiness and fatigue than morning types. In a study of subjective sleep quality conducted with high school students, for instance, Gaina et al. found that morning types were more satisfied with their sleep than evening types [55]. Recently, Dong and colleagues [56] demonstrated that eveningness had a significant prediction effect on poor sleep quality, which was consistent with previous suggestions, remarking that it may be related to the intrinsic properties of the circadian system. These findings support our result that eveningness had significantly poorer subjective sleep quality. Meanwhile, our result also provided empirical evidence for the sleep–wake regulation theory [14], which suggests that disturbed endogenous circadian rhythms (e.g., irregular sleep/wake cycle) can progress to sleep and sleep-related problems such as sleep insufficiency, declined sleep quality and excessive daytime sleepiness. Moreover, according to the social jetlag theory, a delayed sleep phase makes it difficult for individuals to get to school or work on time; hence, those with an evening preference have to reduce their sleep duration on weekdays [57], which may cause disadvantageous subjective mood and sleep in eveningness [58]. Several researchers also found that evening types seem to have more irregular sleep/wake habits and poor voluntary control of sleep habits or inadequate circadian entrainment [59,60], which in turn contributes to worse sleep quality for both youth and adults [40]. Morning types, in contrast, were observed to show the highest level of sleep quality, followed by the intermediate types, which were in line with previous literatures [16,61]. These results suggest that morningness tends to be a protective factor in protecting college students from sleep and psychological problems, whereas if students get used to being associated with an irregular sleep–wake schedule like later bed- and waking-up times, they may be at the risk of suffering from poor sleep quality and sleep-related problems.

Furthermore, we found that bedtime procrastination mediated the negative relation between chronotype and sleep quality, which was consistent with our hypothesis. That is to say, eveningness tends to cause higher levels of bedtime procrastination, which in turn results in poor sleep quality. The observation that chronotype has a significant impact on bedtime procrastination extends the previous work by Kroese et al. [18] and helps to better understand the phenomenon of bedtime procrastination. Research has shown that procrastination, including bedtime procrastination, seems to be a failure of self-regulation caused by poor mechanisms of behavior control [62]. Individuals with low self-regulation can easily give in to temptation and are more susceptible to distractions [29]. Besides, compared with morning types, evening types were found to have poor self-control skills [12], and were more likely to be impulsive and risk-taking, seeking immediate pleasures [25]. Thus, based on previous findings, it is plausible that college students with evening chronotypes were more likely unable to resist temptations like surfing the web, using their smartphones or playing computer games before going to bed, which in turn leads to the delay of planned sleep time, and hence causes sleep insufficiency. Recently, Zhang and Wu further confirmed the mediating role of self-control and bedtime procrastination in the association between chronotype and sleep quality by indicating that eveningness leads to worse self-regulation skills and more bedtime procrastination, which, in turn, is linked to lower levels of sleep quality in young people [28]. Thus, as a bridge between chronotype and sleep quality, bedtime procrastination can be a risk factor to induce poor sleep in college students, especially for those with evening chronotypes. Some intervention and prevention strategies aiming to reduce students’ bedtime procrastination behaviors (e.g., time management, priority-setting and self-regulation skills, mental contrasting with implementation intentions [63,64]) could be taken into daily activities to help them mitigate the negative impact of evening preference on poor sleep quality.

The moderated mediation model showed that sleep hygiene awareness moderated the direct relation between chronotype and sleep quality among college students. This finding partially supported our hypotheses. Specifically, students low in sleep hygiene awareness were inclined to report poorer sleep quality with the increasing level of evening preference, while individuals high in sleep hygiene awareness were inclined to report a lower level of poor sleep quality with the increasing level of evening preference; however, the negative association is stronger for individuals low in sleep hygiene awareness than those high in sleep hygiene awareness (Figure 2). It is likely that individuals with less sleep hygiene knowledge were prone to fall into bad sleep habits such as strenuous exercise, alcohol and caffeine intake before bedtime, the regular use of sleep medication, taking naps during daytime and so forth, which could further exacerbate sleep-related problems including delayed sleep-phase and poor quality of sleep [65], but for those with high levels of sleep hygiene awareness, they could be highly aware of which behaviors are beneficial to sleep and which are not to avoid adverse ones to ensure high-quality nighttime sleep. Further, empirical research has demonstrated that people who prefer to work or stay up late at night need to pay more attention to prevent various kinds of poor sleeping habits before going to bed that induce sleep-related problems [66,67]. Therefore, it may be necessary to develop sleep hygiene education programs as an intervention strategy to improve students’ knowledge of the importance of adopting healthy hygiene practices for better sleep quality and mental health.

Additionally, sleep hygiene awareness also has a moderating effect on the association between chronotype and bedtime procrastination. That is to say, the negative prediction effect of chronotype on bedtime procrastination was significantly stronger in students with higher levels of sleep hygiene awareness (Figure 3). In other words, eveningness was associated with a higher level of bedtime procrastination among those with better sleep hygiene awareness. Since no researcher has ever conducted an examination on the relation between individuals’ chronotype, bedtime procrastination and sleep hygiene awareness, for the current results, we would boldly speculate that although people with better sleep hygiene knowledge are fully aware that going to bed late or postponing bedtimes is not a good habit for nocturnal sleep, they need to go to bed early so as to obtain sufficient sleep due to societal obligations (e.g., work and school schedules) [17], but for those with evening chronotypes, it seems to be difficult to fall asleep in the time frame before the biologically preferred bedtime, indicating that once eveningness generates the intention to go to bed earlier or ‘on time’ to get sufficient sleep, they are prone to failing to do so because biological processes do not support the realization of their intention, and are hence easily led to negative effects such as depression, anxiety or mood swings. Some empirical studies have revealed that negative emotions have a significant positive prediction on individuals’ bedtime procrastination behaviors [68,69]. That is, the more negative moods subjects’ experience, the higher bedtime procrastination they may show. Sirois and colleagues further confirmed from a cross-sectional study that bedtime procrastination was more likely an inadequate emotion regulation strategy to regulate individuals’ immediate negative affect, and thus bedtime procrastinators tend to cope with their negative mood by delaying going to bed, which may, in turn, make them increasingly likely to engage in bedtime procrastination [70].

We did not find that the moderating effect of sleep hygiene awareness in the process of bedtime procrastination affects sleep quality. That is to say, students’ bedtime procrastination could independently predict poor sleep quality, no matter how much sleep hygiene knowledge they obtained. A possible explanation is that as one of the typical poor sleep–wake behaviors, delaying bedtime at night will undoubtedly lead to shorter sleep duration and sleep insufficiency, which in turn causes a decline in the quality of sleep. Not getting enough sleep has also been confirmed by numerous studies to be associated with many other health problems, including lower life satisfaction [71], high levels of anxiety and depression [72], poor academic performance and daytime sleepiness [73]. This result reinforces the importance of keeping regular and healthy hours to ensure a better sleep quality.

## 5. Limitations and Future Directions

The study expanded our understanding of the mechanisms underlying the relationships between circadian typology and physical health, while it also provided a more comprehensive and scientific empirical evidence for the design of intervention programs to reduce bedtime procrastination and foster better sleep hygiene and quality in college students. However, this study had several limitations that should be noted. First, a cross-sectional study design limited the causal effects inference; longitudinal studies are warranted to verify this association in the future. Second, we mainly focused on university students in this work, more studies are needed to further explore whether the results are suitable to other populations such as children and adolescents. Third, our data collection instrument relied on self-report by responding students, which may be susceptible to information bias. Thus, multiple assessments of the studied variables could be adopted to reduce potential measurement bias in future research.

## 6. Conclusions

In summary, the current study examined bedtime procrastination as a mediator and sleep hygiene awareness as a moderator to explore how and when the chronotype is related to sleep quality in Chinese college students. The main conclusions were listed as follows: (1) chronotype was significantly and negatively associated with sleep quality among college students; (2) bedtime procrastination partially mediated the relationship between chronotype and sleep quality; and (3) sleep hygiene awareness had a moderating effect on the direct path of chronotype affecting students’ sleep quality, and it also had a moderating effect in the first half path of the mediating effect of bedtime procrastination on chronotype and sleep quality.

## Figures and Tables

**Figure 1 ijerph-20-00197-f001:**
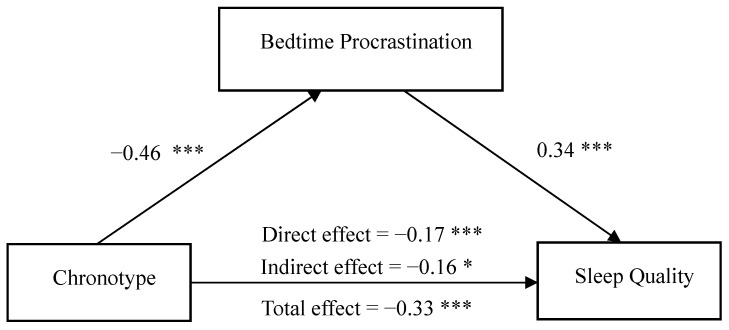
The mediating effect of bedtime procrastination on the relationship between chronotype and sleep quality. * *p* < 0.05, *** *p* < 0.001.

**Figure 2 ijerph-20-00197-f002:**
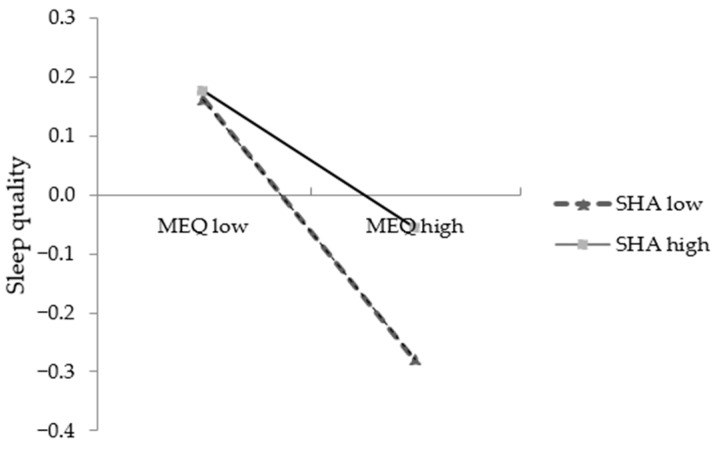
The moderating effect of SHA on the association between MEQ and sleep quality. Note: MEQ: chronotype, SHA: sleep hygiene awareness.

**Figure 3 ijerph-20-00197-f003:**
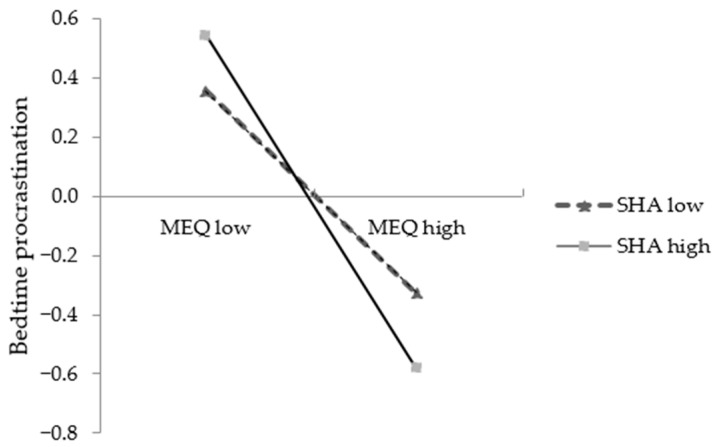
The moderating effect of SHA on the association between MEQ and bedtime procrastination. Note: MEQ: chronotype, SHA: sleep hygiene awareness.

**Table 1 ijerph-20-00197-t001:** Sociodemographic characteristics of participants (*n* = 2822).

Characteristics	*n* (%)
Gender	
Male	808 (28.6%)
Female	2014 (71.4%)
Age	
<20	1311 (46.5%)
≥20	1511 (53.5%)
Grade	
Freshman	866 (30.7%)
Sophomore	1108 (39.2%)
Junior	629 (22.3%)
Senior	180 (6.4%)
Postgraduate	39 (1.4%)
Chronotype	
Morning-type	765 (27.1%)
Intermediate-type	1880 (66.6%)
Evening-type	177 (6.3%)

**Table 2 ijerph-20-00197-t002:** Descriptive statistics for the bedtime procrastination, sleep quality and sleep hygiene awareness in the total sample according to chronotype.

Variables	Morning-Type	Intermediate-Type	Evening-Type	*F*	*p*-Value
Bedtime procrastination	2.56 ± 0.62	3.00 ± 0.60	3.61 ± 0.61	254.67	<0.001
PSQI total score	4.53 ± 2.64	5.99 ± 2.73	7.56 ± 3.20	116.37	<0.001
Subjective sleep quality	0.83 ± 0.65	1.16 ± 0.71	1.41 ± 0.82	81.11	<0.001
Sleep disturbance	0.84 ± 0.58	0.98 ± 0.55	1.08 ± 0.60	20.69	<0.001
Sleep efficiency	0.41 ± 0.72	0.51 ± 0.81	0.69 ± 0.97	9.63	<0.001
Sleep latency	1.03 ± 0.87	1.35 ± 0.95	1.71 ± 1.08	45.93	<0.001
Sleep duration	0.45 ± 0.54	0.58 ± 0.56	0.66 ± 0.67	18.83	<0.001
Hypnotic medication	0.03 ± 0.25	0.05 ± 0.29	0.12 ± 0.49	6.59	<0.01
Daytime dysfunction	0.94 ± 0.85	1.36 ± 0.88	1.88 ± 0.95	101.46	<0.001
Sleep hygiene awareness	55.80 ± 10.55	55.71 ± 8.57	56.10 ± 7.14	0.16	0.849

Notes: Estimated marginal means were adjusted for covariates (mean ± standard deviation). Post-hoc comparisons among different chronotype groups were performed with adjusted values.

**Table 3 ijerph-20-00197-t003:** Mean, standard deviation and correlations between main variables (*n* = 2822).

Variables	M	SD	1	2	3	4	5
1. Gender	—	—	—				
2. Age	19.77	1.41	0.12 ***	—			
3. MEQ	53.95	7.65	0.09 ***	0.01	—		
4. BPS	2.92	0.66	−0.08 ***	0.01	−0.47 ***	—	
5. PSQI	5.69	2.85	−0.07 ***	0.06 **	−0.33 ***	0.42 ***	—
6. SHA	55.76	9.08	−0.09 ***	−0.24	−0.02	−0.03	0.06 **

Notes: MEQ: Morningness–Eveningness Questionnaire; BPS: Bedtime Procrastination Scale; PSQI: Pitts burg Sleep Quality Index; SHA: Sleep Hygiene Awareness. ** *p* < 0.01. *** *p* < 0.001.

**Table 4 ijerph-20-00197-t004:** The mediation effect of bedtime procrastination on the association between chronotype and sleep quality.

Predictors	Model 1 (PSQI)			Model 2 (BPS)			Model 3 (PSQI)		
	*β*	*t*	95% CI	*β*	*t*	95% CI	*β*	*t*	95% CI
Gender	−0.04	−2.48 *	[−0.18, −0.20]	−0.04	−2.18 *	[−0.15, −0.01]	−0.03	−1.88	[−0.14, 0.00]
Age	0.07	3.98 ***	[0.03, 0.08]	0.02	1.37	[−0.01, 0.04]	0.06	3.74 ***	[0.02, 0.07]
MEQ	−0.33	−18.28 ***	[−0.36, −0.29]	−0.46	−27.60 ***	[−0.50, −0.43]	−0.17	−8.89 ***	[−0.21, −0.13]
BPS							0.34	17.79 ***	[0.30, 0.38]
*R* ^2^	0.11 121.18 ***			0.22 261.61 ***			0.20 180.16 ***		
*F*									

Notes: MEQ: Morningness–Eveningness Questionnaire; BPS: Bedtime Procrastination Scale; PSQI: Pittsburg Sleep Quality Index. Male = 1, Female = 0. *β*: Standardized coefficients. * *p* < 0.05, *** *p* < 0.001.

**Table 5 ijerph-20-00197-t005:** The moderated mediation effects of chronotype on participants’ sleep quality.

Variables	Model 1 (Outcome: BPS)				Model 2 (Outcome: PSQI)			
	*β*	*SE*	*t*	95% CI	*β*	*SE*	*t*	95% CI
Gender	−0.04	0.04	−2.54 *	[−0.08, −0.01]	−0.02	0.04	−1.43	[−0.06, 0.01]
Age	0.02	0.01	1.44	[−0.01, 0.06]	0.06	0.01	3.76 ***	[0.03, 0.10]
MEQ	−0.45	0.02	−27.15 ***	[−0.49, −0.42]	−0.17	0.02	−8.87 ***	[−0.21, −0.13]
SHA	−0.02	0.02	−0.97	[−0.05, 0.02]	0.06	0.02	3.41 **	[0.03, 0.09]
MEQ × SHA	−0.11	0.02	−6.57 ***	[−0.14, −0.08]	0.05	0.02	2.73 **	[0.02, 0.09]
BPS					0.35	0.02	17.30 ***	[0.31, 0.39]
BPS × SHA					0.01	0.02	0.36	[−0.03, 0.05]
*R* ^2^	0.23 169.62 ***				0.21 107.47 ***			
*F*								

Notes: MEQ: Morningness–Eveningness Questionnaire; BPS: Bedtime Procrastination Scale; PSQI: Pittsburg Sleep Quality Index. SHA: Sleep Hygiene Awareness. Male = 1, Female = 0. *β*: Standardized coefficients. * *p* < 0.05; ** *p* < 0.01; *** *p* < 0.001.

## Data Availability

The datasets generated during and/or analyzed during the current study are available from the corresponding author on reasonable request.

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
