# Peer review of "Association between Chronotype and Sleep Quality among Chinese College Students: The Role of Bedtime Procrastination and Sleep Hygiene Awareness"

_ijerph, 2022, doi:10.3390/ijerph20010197_

Round 1

Reviewer 1 Report

This study investigated the association between chronotype and sleep quality, as well as the mediating effect of bedtime procrastination and the moderating effect of sleep hygiene awareness in Chinese college students, which comprehensively examined the relationship between different sleep variables. However, there are still some points that needed to be improved.

Abstract

1.      The beginning of the Abstract directly describes the research purpose of this study, without a certain research background introduction. It’s better to briefly introduce the research background at the beginning of the abstract, and then describe the purpose of this study.

2.      Please indicate the detailed subject information in brackets after “A sample of 2822 college students”, including the number of students, mean age, SD, and gender ratio, etc.

3.      The last sentence of abstract “The theoretical and practical implications of these results are discussed”, is it appropriate to place this sentence in the conclusion part of the abstract? In other way, the author should list some research implications in detail here.

Introduction

1.      It is suggested that in addition to the definition of chronotype, the author should also include the explanation of the concept and definition of sleep quality in the introduction section, and then clarify the relationship between these two variables.

2.      In paragraph 2, line 48-49, it stated that “Research has shown that…among evening types”. However, the author only made a general hypothesis about the relationship between “chronotype” and sleep quality when proposing the hypothesis (line 63-64). If the three dimensions of chronotype were separated, it would be more reasonable. Also, the relationship between evening-types and sleep quality should be specially emphasized.

3.      The authors should have added more references on the relationship between different chronotypes and sleep quality to support the hypothesis, rather than stating too much about the relationship between chronotypes and other sleep problems.

4.      Is there any theory to support the relationship between chronotype and sleep quality? It would be better to cite some theories.

5.      The two paragraphs in 1.1 should be reversed, firstly clarify the predictive effect of chronotype on bedtime procrastination, secondly the predictive effect of bedtime procrastination on sleep quality.

Material and Methods

1.      How is the sample selected for this study? Why are the percentages of different grades so different? Please describe the sampling method in this section.

2.      Much more detail on the sample construction is needed in Section “2.1. Participants and Procedure”. For example, where did the participants come from? Since they are Chinese college students, which district of China are they from? And which colleges or universities are they from? This information should all be clarified in this section.

3.      What are the sociodemographic characteristics of the participants? It is suggested that the author add a table to present the data.

4.      Label the four scales as 2.2.1, 2.2.2... will be more clear and organized.

5.      In line 183-184, “Items have a response scale with four or five options”, which questions have four options? And which have five options? What is the detail information? Please clarify these questions.

6.      Please cite relevant references to illustrate the reliability and validity of these scales.

7.      The description of Section “2.3 Statistical Analysis” is rather confusing, so it is suggested to explain it step by step.

Results

1.      In Section 3.1, information about gender is best presented in the Methods section.

2.      I suggest that the author also label indirect effect values in Table 3 or Figure 2, and include “*p<0.05” in the note of Table 3, because “ * ” appears in the table.

Discussion

1.      In line 344-345, “and also the indirect path of …”, the results showed that SHA only moderated the path between chronotype and bedtime procrastination, so it may be not plausible to directly describe it as “indirect path” here.

2.      In the discussion section, the author focused on the influence of evening-type on sleep quality, but ignored the morning-type and intermediate-type. The results of the other two types should also be discussed.

3.      The explanation of the research results lacks the support of relevant theories, so some theories should be provided to support the results.

Limitations and Future Directions

1.      The author should briefly state the strengths of this study.

2.      Part of the research limitation did not present the corresponding future research direction, which was too concise.

3.      The last limitation of the study seems unreasonable and is suggested to be deleted. It is absurd for the author to take "only the relationships between these four variables (chronotype, bedtime procrastination, sleep quality and sleep hygiene awareness) were analyzed" as a limitation. This study already investigated the relationship between these four variables, if this is taken as a limitation of this study, it seems to negate the entire existing research, which is a paradoxical statement.

4.      The prevention and intervention measures mentioned in this study could be more detailed.

Reviewer 2 Report

The paper is well written and executed, please check for typos errors, and grammatical mistakes.

Is it necessary to give subheadings in the introduction? figure 1 can better be removed from this section

Concise the conclusion, focus on the main achievement as per the objective of the study, and any other information can be added in the discussion section.

Reviewer 3 Report

What was the power of the study?

What version of Bedtime Procrastination Scale was used? English or Chinese one? The authors failed to mention that detail.
